Effects of maximum residue limit of triflumezopyrim exposure on fitness of the red imported fire ant Solenopsis invicta

Li Qiting 1
Zhao Fei 1
Li Jiayi 1
Tao QiuHong 1
Gao JiaQian 2
Lu Yong-Yue 1
Wang Lei 1 wanglei1107@outlook.com
1 College of Agriculture, South China Agricultural University , Guangzhou, Guangdong , China
2 Guangdong Tianhe Agricultural Means of Production Co., Ltd. , Guangzhou, Guangdong , China
Gillespie Joseph
Electronic publication date: 2019 Dec 10
Publication date: 2019
Volume: 7
Electronic Location ID: e8241
Received 2019 Jul 12; Accepted 2019 Nov 19
Copyright: © 2019 Li et al.
Copyright year: 2019
Copyright holder: Li et al.
License: This is an open access article distributed under the terms of the Creative Commons Attribution License, which permits unrestricted use, distribution, reproduction and adaptation in any medium and for any purpose provided that it is properly attributed. For attribution, the original author(s), title, publication source (PeerJ) and either DOI or URL of the article must be cited.
License URL: https://creativecommons.org/licenses/by/4.0/

Keywords: Triflumezopyrim, Fire ant, nAChR, Aggressiveness, Colony growth, Behavior

Funding: Natural Science Foundation of Guangdong Province 2018A030313823 Agricultural Science and Technology Innovation and Extension Project of Guangdong Province 2018LM2161 This project was supported by the Natural Science Foundation of Guangdong Province (Grant No: 2018A030313823), and the Agricultural Science and Technology Innovation and Extension Project of Guangdong Province (Grant No: 2018LM2161). The funders had no role in study design, data collection and analysis, decision to publish, or preparation of the manuscript.

==============================
The impact of exposure to free feeding concentrations of triflumezopyrim to the red imported fire ant, Solenopsis invicta, in maximum residue tolerances for 56 days was investigated to understand whether triflumezopyrim, a novel neonicotinoid, poses unacceptable risks to the environment. Our results demonstrated that neither 0.5 μg/ml nor 0.2 μg/ml triflumezopyrim have a significant impact on the growth of the S. invicta colony and their food consumption (sugar water and locusts) during the length of treatment. While both 0.5 μg/ml and 0.2 μg/ml triflumezopyrim improved the grasping ability of S. invicta, and 0.5 μg/ml not 0.2 μg/ml triflumezopyrim increased their rate of locomotion. In addition, although 0.5 μg/ml and 0.2 μg/ml triflumezopyrim increased their individual aggressiveness index, the probability of the survival of S. invicta was not impacted by triflumezopyrim treatments in aggressive group encounters. This study suggests that triflumezopyrim did not have a negative impact on the fitness of S. invicta at 0.5 μg/ml and 0.2 μg/ml exposures.

Introduction

Neonicotinoids are insecticides that act on the nicotinic acetylcholine receptor of insects; it exhibits good activity on sent hopper controap-sucking pests and comprises over 30% of insecticide sales in the world (Hladik, Main & Goulson, 2018; Ihara & Matsuda, 2018). However, the negative impact of neonicotinoids on non-target organisms at their sublethal concentrations is of increasing concern (Butcherine et al., 2019; Wang et al., 2018). Research has shown that neonicotinoids have adverse impacts on pollinator insects, especially honeybees, and three types of this chemical are currently banned from the European Union, including imidacloprid, thiamethoxam and clothianidin (Ihara & Matsuda, 2018; Pisa et al., 2015).

Current, novel neonicotinoids were developed including sulfoxaflor, flupyradifurone and triflumezopyrim. Researchers have considered the safety of these new neonicotinoids over other neonicotinoids for non-target insects, especially honeybees. Pan, Lu & Wang (2017) found that 1.0 μg/ml sulfoxaflor reduced aggressiveness, colony growth, and food consumption of the fire ant Solenopsis invicta, while Siviter et al. (2019) found that acute sulfoxaflor exposure did not have negative impacts the olfactory conditioning and working memory of bees. Research showed that flupyradifurone impairs olfactory learning, motor abilities, and the survival of honeybees, including Apis mellifera carnica, A. mellifera ligustica, and A. cerana at even field-realistic doses (Hesselbach & Scheiner, 2019; Tan et al., 2017; Tosi & Nieh, 2019).

Triflumezopyrim is developed by DuPont Crop Protection and is effective against sap-sucking pests (Cordova et al., 2016; Zhang, 2017; Zhang et al., 2017). It was registered in China, Korea, and the Philippines in 2016 and Pyraxalt™ 10% triflumezopyrim SC, the commercial product, was brought to market in China in 2017 (http://news.agropages.com/News/NewsDetail---19109.htm, access on 2019-6-2; http://www.agroinfo.com.cn/other_detail_4668.html, access on 2019-6-2). Zhu et al. (2018) found that triflumezopyrim is an effective insecticide for suppressing plant hopper populations and is harmless to their natural enemies including Anagrus nilaparvatae, Cyrtorhinus lividipennis, Paederus fuscipes, Pirata subpiraticus, Ummeliata insecticeps, Hylyphantes graminicola, and Pardosa pseudoannulata. Since sulfoxaflor and flupyradifurone were reported to have negative effects on non-target insects, the effects of triflumezopyrim on non-target insects deserves greater attention.

Ants are an important species and a crucial member of their ecosystem; their disruption would negatively impact the health of the ecosystem (Toro, Ribbons & Pelini, 2012). Imidacloprid can change the foraging and competitive behaviors of Lasius niger and L. flavus at sublethal concentrations, which may alter the structure and dynamics of ant communities after oral treatment (Thiel & Köhler, 2016). One μg/ml imidacloprid oral treatment reduced the aggressive behavior of the native ant, Monomorium antarcticum, to the invasive ant, Linepithema humile, which may facilitate a successful invasion by the invasive species (Barbieri et al., 2013). Ants are important predators and terrestrial herbivores with a large amount of soil contact and were exposed to neonicotinoids by oral exposure through plant ingestion as well as by contact exposure through contaminated soil (Hölldobler & Wilson, 1990).

The red imported fire ant, S. invicta, is a well-studied ant species making it an ideal model on which to study the sublethal effects of neonicotinoids on ants (Vinson, 2013; Wurm et al., 2011). S. invicta is an active and efficient predator of pests in agroecosystems despite its negative effects on public health, agriculture and biodiversity (Harvey & Eubanks, 2004; Vogt et al., 2001; Wang et al., 2016). In this study, we exposed S. invicta colonies to residue limits of triflumezopyrim and evaluated the impact of exposure on their fitness. The objective of the study was to determine the effects of the maximum residue limit of triflumezopyrim on the fitness of the colony and the behavior of workers.

Materials and Methods

Insects and insecticide

Red imported fire ants, Solenopsis invicta, were collected from an organic farm in Conghua County, Guangzhou, China and reared at the Red Imported Fire Ant Research Centre at the South China Agricultural University. S. invicta colonies were provided with a 10% w/w sugar water solution and frozen locusts Locusta migratoria. The social form of S. invicta colonies were determined using methods of Shoemaker & Ascunce (2010), and only polygyne colonies were used in this experiment. All bioassays were conducted under laboratory conditions, and all colonies were reared in the laboratory at least 1 week prior to the start of the experiment.

The commercially available insecticide, Pyraxalt™ 10% triflumezopyrim SC (DuPont™ Crop Protection, Pudong, Shanghai, China), was used in all experiments.

Effect of triflumezopyrim on S. invicta food consumption and colony growth

The Environmental Protection Agency (EPA) of the United States establishes tolerances for the residues of triflumezopyrim in rice grains and rice hulls of 0.4 ppm and 1.0 ppm, respectively (https://www.federalregister.gov/documents/2017/10/16/2017-22356/triflumezopyrim-pesticide-tolerances, accessed on 2019-6-4). Therefore, we used 0.5 and 0.2 μg triflumezopyrim/ml for the residue tolerances of triflumezopyrim based on the regulatory requirements of the EPA.

Pyraxalt™ is advertised to provide up to 25 days of consistent hopper control (http://www.dupont.com/products-and-services/crop-protection/pyraxalt-insect-control.html, access on 2019-6-2). The estimated average life span of minor, medium, and major workers of S. invicta is 45 days, 75 days and 135 days, respectively (Calabi & Porter, 1989). In light of these two combined factors the experiment was conducted over 8 weeks.

The experimental method was reflective of the work of Pan, Lu & Wang (2017) in which a colony was divided into the same weight (>20 g), and each sub-colony contained workers, brood, larva, pupa, and at least one functional queen. Each sub-colony then received either the control 10% w/w sugar water or a single concentration of triflumezopyrim (0.2 μg/ml or 0.5 μg/ml) in 10% w/w sugar water; solutions were provided in a 15 ml glass tubes with a cotton stopple. Sub-colonies received a sufficient amount of locusts. The weight of each sub-colony was taken every 7 days to evaluate the effect of triflumezopyrim on colony growth of S. invicta. The accumulated percentage weight loss of each sub-colony was calculated after 7 days, 14 days, 21 days, 28 days, 35 days, 42 days, 49 days and 56 days of treatment (Pan, Lu & Wang, 2017). Sugar water and locust consumption (mg sugar water or locust per g ant) was also calculated for each sub-colony at 1 day, 7 days, 14 days, 21 days, 28 days, 35 days, 42 days, 49 days and 56 days. Water evaporation was also estimated. Four colonies were used in this experiment.

The effect of triflumezopyrim on active behavior of S. invicta

We evaluated the effect of triflumezopyrim on S. invicta behavior by measuring their grasping ability and walking speed after 56 days of treatment. The experiment method was reflective of the method used by Huang et al. (2016) and Wang, Zeng & Chen (2015).

To test the grasping ability of the fire ant, 10 workers varying in size from the treatment or control sub-colony were placed on an A4-sized paper. The paper was gently turned over 180 degrees after 10 s and held in place for 10 s. Workers falling from the paper was considered to have lost their grasping ability. The grasping ability was evaluated by grasping rate using the following formula: grasping rate (%) = (number of workers possessing grasping ability/total number of workers in test) × 100%. Each sub-colony from each treatment and control group was tested three times.

To test the walking speed of the fire ants, a tray with a fire ant sub-colony was connected to a forging arena (tray, L × W × H, 41.75 × 27.5 × 12 cm) by a wood bridge. The bridge consists of two 20 cm vertical segments and one 20 cm horizontal segment. The walking speed of the workers was measured by recording the time needed for an ant to walk across the 20 cm horizontal segment of the bridge. The method was described by Wang, Zeng & Chen (2015). A total of 25 workers were measured from each sub-colony.

The effect of triflumezopyrim on the aggressiveness of S. invicta

In this experiment, we assessed the effect of sublethal doses of triflumezopyrim on the aggressive behavior of S. invicta. The experimental method was reflective of the method used by Pan, Lu & Wang (2017). The aggressiveness level of S. invicta was assessed when S. invicta individuals and colonies were confronted with another ant species, Pheidole fervida, at 56 days after treatment. Four P. fervida colonies were used.

S. invicta and P. fervida colonies were paired in inter specific individual or colony interaction tests. The inter specific interaction was quantified using a behavioral assay described by Pan, Lu & Wang (2017).

In inter specific individual aggression, interactions were scored and calculated as described by Rice & Silverman (2013) and Pan, Lu & Wang (2017). Four replicate experiments were performed for three pairs of colonies with each receiving a dose of triflumezopyrim. Each replicate involved different workers.

Inter specific colony aggression was evaluated using an assay described by Pan, Lu & Wang (2017). Specially, 30 workers varying in size from pairs of treated S. invicta and untreated P. fervida colonies were randomly chosen and introduced into a petri dish. Mortality rates of S. invicta were recorded after 3 h. Three replicates were performed for three pairs of colonies receiving a dose of triflumezopyrim at the same time. Each replicate involved different workers.

Statistical analyses

All data were analyzed using Shapiro–Wilk and Levene’s tests for normal distribution and homogeneity of variances, respectively. If data were normally distributed and had similar variances, then the means of measured variables were compared by one-way analysis of variance (ANOVA). Significant ANOVA results were corrected for multiple comparisons using LSD’s method. Non-normally distributed data were analyzed using a non-parametric Kruskal–Wallis test to compare medians; differences significant at the 0.05 significance level were subjected to a Mann–Whitney test for pairwise comparisons. All statistical analyses were performed using SPSS version 18.0 (SPSS Inc., Chicago, IL, USA).

Results

Effect of triflumezopyrim on S. invicta colony growth

There is no significant difference in the survival of S. invicta colonies among 0.5 μg/ml triflumezopyrim, 0.2 μg/ml triflumezopyrim, and the control (Fig. 1; day 7, LSD test, F2, 9 = 0.404, P = 0.679; day 14, LSD test, F2, 9 = 0.009, P = 0.991; day 21, LSD test, F2, 9 = 0.116, P = 0.892; day 28, LSD test, F2, 9 = 0.089, P = 0.916; day 35, LSD test, F2, 9 = 0.298, P = 0.749; day 42, LSD test, F2, 9 = 0.746, P = 0.501; day 49, LSD test, F2, 9 = 0.488, P = 0.629; day 56, LSD test, F2, 9 = 0.527, P = 0.608). After 56 days of treatment, the accumulated percentage weight losses were 51.40%, 50.65% and 58.46% in colonies treated with 0.5 μg/ml triflumezopyrim, 0.2 μg/ml triflumezopyrim, and controls, respectively.

Figure 1 Accumulated colony weight loss (mean percentage ± SE) over 56 days after treatment of Solenopsis invicta with diﬀerent concentrations of triflumezopyrim.

Effect of triflumezopyrim on S. invicta food consumption

There is no significant difference in sugar consumption between ants treated with 0.5 μg/ml triflumezopyrim, 0.2 μg/ml triflumezopyrim, and the control (Table 1; day 1, LSD test, F2, 9 = 0.946, P = 0.418; day 7, LSD test, F2, 9 = 0.673, P = 0.534; day 14, LSD test, F2, 9 = 0.322, P = 0.733; day 21, LSD test, F2, 9 = 1.396, P = 0.296; day 28, LSD test, F2, 9 = 2.129, P = 0.175; day 35, LSD test, F2, 9 = 0.283, P = 0.760; day 42, LSD test, F2, 9 = 1.472, P = 0.280; day 49, LSD test, F2, 9 = 1.768, P = 0.225; day 56, LSD test, F2, 9 = 0.782, P = 0.486).

Table 1 Sugar water consumption (mean ± SE) after 56 days treatment (number of colonies/treatment group = 4).

Day after treatment	Sugar water consumption (mg per gram workers)	Results of the one-way ANOVA	
Control	0.2 μg/ml	0.5 μg/ml	F2, 9	P	
1	149.7 ± 37.3a	214.6 ± 36.8a	195.5 ± 26.9a	0.946	0.418	
7	145.4 ± 29.4a	120.5 ± 25.6a	102.6 ± 23.2a	0.673	0.534	
14	139.7 ± 11.5a	152.1 ± 25.3a	130.8 ± 17.2a	0.322	0.733	
21	153.5 ± 41.2a	185.7 ± 19.0a	112.6 ± 28.7a	1.396	0.296	
28	135.8 ± 30.7a	77.0 ± 12.9a	96.9±12.3a	2.129	0.175	
35	145.9 ± 17.3a	129.4 ± 22.3a	154.8 ± 31.0a	0.283	0.760	
42	95.9 ± 16.7a	66.4 ± 7.9a	90.1 ± 12.5a	1.472	0.280	
49	84.5 ± 8.4a	139.8 ± 34.8a	100.8 ± 9.5a	1.768	0.225	
56	81.2 ± 10.8a	99.8 ± 11.9a	98.3 ± 12.2a	0.782	0.486	
Note:

Same letter represents no significant difference within each observation period (P > 0.05).

Triflumezopyrim treatments do not have an impact on locust consumption (Table 2; day 1, LSD test, F2, 9 = 0.004, P = 0.996; day 7, LSD test, F2, 9 = 0.172, P = 0.845; day 14, LSD test, F2, 9 = 0.539, P = 0.601; day 21, LSD test, F2, 9 = 1.726, P = 0.232; day 28, LSD test, F2, 9 = 1.443, P = 0.286; day 35, LSD test, F2,9 = 3.691, P = 0.068; day 42, LSD test, F2, 9 = 0.728, P = 0.509; day 49, LSD test, F2, 9 = 1.940, P = 0.199; day 56, LSD test, F2, 9 = 0.089, P = 0.915).

Table 2 Locust consumption (mean ± SE) after 56 days treatment (number of colonies/treatment group = 4).

Day after treatment	Sugar water consumption (mg per gram workers)	Results of the one-way ANOVA	
Control	0.2 μg/ml	0.5 μg/ml	F2, 9	P	
1	71.4 ± 47.0a	76.5 ± 33.3a	72.6 ± 47.6a	0.004	0.996	
7	33.5 ± 33.4a	41.9 ± 18.6a	20.0 ± 26.1a	0.172	0.845	
14	17.5 ± 13.9a	38.0 ± 16.8a	30.3 ± 10.9a	0.539	0.601	
21	30.6 ± 8.4a	63.6 ± 20.2a	67.1 ± 15.1a	1.726	0.232	
28	23.8 ± 7.5a	33.9 ± 5.9a	15.0 ± 9.8a	1.443	0.286	
35	39.6 ± 16.6a	67.8 ± 18.0a	102.8 ± 14.5a	3.691	0.068	
42	−15.1 ± 19.9a	10.3 ± 23.1a	22.4 ± 24.1a	0.728	0.509	
49	−1.6 ± 12.1a	49.6 ± 29.1a	61.6 ± 27.3a	1.940	0.199	
56	36.6 ± 17.4a	40.4 ± 28.2a	52.8 ± 36.0a	0.089	0.915	
Note:

Same letter represents no significant difference within each observation period (P > 0.05).

Effect of triflumezopyrim on S. invicta activity behavior

After 56 days of treatment the grasping rate of workers was 88.67 ± 2.14%, 90.33 ± 1.67% and 71.67 ± 5.13% in 0.5 μg/ml triflumezopyrim, 0.2 μg/ml triflumezopyrim, and the control, respectively. The grasping rate of workers in the triflumezopyrim treatment is higher than that of the control (0.5 μg/ml triflumezopyrim and control, Mann–Whitney test, U = 27, P = 0.009; 0.2 μg/ml triflumezopyrim and control, Mann–Whitney test, U = 18, P = 0.002).

After 56 days of treatment the walking speed of workers was 9.50 ± 0.22 mm/s, 9.39 ± 0.23 mm/s and 9.41 ± 0.38 mm/s in 0.5 μg/ml triflumezopyrim, 0.2 μg/ml triflumezopyrim, and control, respectively. The walking speed of workers in 0.5 μg/ml triflumezopyrim treatment was higher than that of the control (0.5 μg/ml triflumezopyrim and control, Mann–Whitney test, U = 4044.5, P = 0.02), and the walking speed of workers in the 0.2 μg/ml triflumezopyrim treatment is not significantly different from either that in the 0.5 μg/ml triflumezopyrim or the control (0.2 μg/ml triflumezopyrim and control, Mann–Whitney test, U = 4199.5, P = 0.050; 0.2 μg/ml triflumezopyrim and 0.5 μg/ml triflumezopyrim, Mann–Whitney test, U = 4733.0, P = 0.514).

Effect of triflumezopyrim on S. invicta aggressiveness

After 56 days of treatment the aggressiveness indices of S. invicta workers was 1.58 ± 0.28, 1.31 ± 0.29 and 0.47 ± 0.20 in 0.5 μg/ml triflumezopyrim, 0.2 μg/ml triflumezopyrim, and the control, respectively. The aggressiveness index of S. invicta was significantly increased by triflumezopyrim treatment (0.5 μg/ml triflumezopyrim and control, Mann–Whitney test, U = 57.5, P = 0.004; 0.2 μg/ml triflumezopyrim and control, Mann–Whitney test, U = 75.5, P = 0.030).

In the group aggression experiment, the mortality of S. invicta workers was 6.33 ± 1.51%, 6.67 ± 1.80%, 7.33 ± 2.14% in 0.5 μg/ml triflumezopyrim, 0.2 μg/ml triflumezopyrim, and control after 56 days of treatment, respectively; there is no significant difference among triflumezopyrim treatments and the control (LSD test, F2, 33 = 0.77, P = 0.926).

Discussion

The negative impact of neonicotinoids on non-target organisms is a global concern. As a novel neonicotinoid, the impact of triflumezopyrim on non-target insects deserves attention. Zhu et al. (2018) found that 62.5 μg·a.i/ml triflumezopyrim does not negatively impact the parasitic wasp Anagrus nilaparvatae after 30 min of exposure. Our study showed that 0.2 μg/ml and 0.5 μg/ml triflumezopyrim, which are near the tolerance levels for the residue of triflumezopyrim in rice grains and hulls, does not have a negative effect on the colony growth and food consumption (sugar water and locusts) of fire ants after 56 day of exposure. Although the 0.2 μg/ml and 0.5 μg/ml triflumezopyrim treatments increased the grasping ability and individual aggressiveness of fire ant workers, and 0.5 μg/ml triflumezopyrim treatment increased the walking speed of fire ants, triflumezopyrim treatment did not impact the mortality of fire ant workers in the group aggression experiment.

Rondeau et al. (2014) suggested that testing for the chronic effects of pesticides should be extended to 30 days or more for social insects. The estimated average life span of minor, medium, and major workers is 45 days, 75 days and 135 days, respectively (Calabi & Porter, 1989). Our experiments were conducted for 56 days in light of the above mentioned factors. We used three experimental parameters to investigate the impact of tolerances for the residues of triflumezopyrim exposure on fire ant colony growth, food consumption and activity. The three parameters were chosen because they are related to the fitness of the fire ant colony. For example, walking speed and aggressiveness is involved in competition for food, and the protection and expansion of territory, which are indicators of competitiveness in ants (Holway, 1999).

Our results showed that the treatments of 0.2 μg/ml and 0.5 μg/ml triflumezopyrim increased the grasping ability and aggressiveness of the individual worker and 0.5 μg/ml triflumezopyrim increased the walking speed of the fire ant. Other neonicotinoids have a similar effect. For example, thiamethoxam at 108.1 ng/g exposure caused a short-term locomotor hyperactivity of the carabid beetle Platynus assimilis (Tooming et al., 2017). The lower concentration of imidacloprid, such as 10 ng or 1.25 ng/insect treatment, resulted in a higher locomotor activity of the treated insect (Galvanho et al., 2013; Lambin et al., 2001). De França et al. (2017) indicated that insect behavior changes with sublethal doses of pesticides, which may indirectly influence species biology and fitness. However, behavior changes resulting in fire ant fitness for colony growth and group aggression was not impacted based on our data.

Registration information of triflumezopyrim is submitted to many key markets of the world and is now registered for sale in China (http://news.agropages.com/News/NewsDetail---19109.htm). Our study revealed that there is no negative impact of 0.5 μg/ml triflumezopyrim exposure on the colony growth and food consumption of the fire ant. Further studies are needed to determine the conditions under which triflumezopyrim may have a detrimental impact on ants. Our results provide information for the future use of this insecticide on crops in terms of regulations and policy decisions.

Conclusions

We found triflumezopyrim, a novel neonicotinoid, did not cause any negative impact on the fitness of the fire ant S. invicta at 0.2 μg/ml and 0.5 μg/ml after 56 days of observation. Neither colony growth nor food consumption was influenced by 0.2 μg/ml and 0.5 μg/ml triflumezopyrim. The probability of survival of S. invicta was not impacted by triflumezopyrim treatments in aggressive group encounters, although 0.5 μg/ml and 0.2 μg/ml triflumezopyrim increased the individual aggressive index, and 0.5 μg/ml triflumezopyrim increased the moving speed and grasping ability of S. invicta. The results imply that the maximum residue limit of triflumezopyrim may have no impact on other non-target ants.

Supplemental Information

Supplemental Information 1 Raw Data of dose residues of triflumezopyrim exposure have negative effect on ant?

Click here for additional data file.

We thank Jiefu Deng and Dong Ning for their valuable help in the experimental setup. We also thank the editor and reviewers for their constructive comments on this paper.

Additional Information and Declarations

Competing Interests

Author Contributions

Data Availability

The authors declare that they have no competing interests.

Qiting Li performed the experiments, prepared figures and/or tables, authored or reviewed drafts of the paper, approved the final draft.

Fei Zhao performed the experiments, prepared figures and/or tables, authored or reviewed drafts of the paper, approved the final draft.

Jiayi Li performed the experiments, prepared figures and/or tables, approved the final draft.

QiuHong Tao performed the experiments, prepared figures and/or tables, approved the final draft.

JiaQian Gao analyzed the data, contributed reagents/materials/analysis tools, authored or reviewed drafts of the paper, approved the final draft.

Yong-Yue Lu conceived and designed the experiments, analyzed the data, contributed reagents/materials/analysis tools, authored or reviewed drafts of the paper, approved the final draft.

Lei Wang conceived and designed the experiments, performed the experiments, analyzed the data, contributed reagents/materials/analysis tools, prepared figures and/or tables, authored or reviewed drafts of the paper, approved the final draft.

The following information was supplied regarding data availability:

Raw data for this manuscript is available in a Supplemental File.

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
