# Peer review of "Effects of maximum residue limit of triflumezopyrim exposure on fitness of the red imported fire ant Solenopsis invicta"

_PeerJ, doi:10.7717/peerj.8241_

## Round 0.1 · original submission · Major Revisions

Dear Dr. Li and colleagues:

Thanks for submitting your manuscript to PeerJ. I have now received two independent reviews of your work, and as you will see, one reviewer raised serious concerns about the research and recommended rejection. The other reviewer was more enthusiastic, those still raised some serious concerns. Thus, there is a lot of criticism here for you to consider. If you choose to address these concerns, I invite you to revise your work and resubmit. However, I strongly encourage you to take into account all of the concerns raised by both reviewers.

It appears that you cite some papers that have been published that suggest similar results with other neonicotinoids against RIFA. It is not clear how these studies reveal anything meaningful about the environmental impact of triflumezopyrim. It is not clear how ants are actually involved. RIFA is not a big honeydew feeding ant, but it might be a predator of the rice pests. However, it is not clear how the ants are consuming the toxicant and how this would justify using triflumezopyrim. Please address this ambiguity.

I look forward to seeing your revision, and thanks again for submitting your work to PeerJ.

Good luck with your revision,

-joe

Reviewer 1 ·

Basic reporting

The manuscript needs a thorough editorial review to eliminate awkward phrases, subject-verb tenses, and singular and plural subjects. The similarity of this paper and Pan et al. (2017) is pretty noticeable. The only significant difference is that triflumezopyrim was tested instead of sulfoxaflor. I am not sure it is really telling us something new about these new neonicotinoids.

Maybe if the authors could explain why and how ants are important nontarget insects of neonicotinoids. I thought the issue with neonicotinoids was with nectar, pollen and honeybees and maybe parasitoids of some plant insects. I don't see what the results of this study tell us about these new neonicotinoids.

"The objectives of the study were to determine the effect of field residues of triflumezopyrim on colony fitness and the behavior of workers." Field residues are actually food tolerance in rice grains and hulls. Are the ants actually feeding on the rice and hulls?

How does RIFA enter into the rice ecosystem? Maybe this could be better explain why they are being tested.

Experimental design

I am having difficulty understanding how these studies tell us anything about the risks of triflumezopyrim in the environment. I don’t see the connection between red imported fire ants (RIFA) survival and non-targets, especially pests. The two concentrations tested had no effect on RIFA. So!!!!

The food tolerances of triflumezopyrin in rice grains and hulls are actually 1.0 μg/ml and 0.4 μg/ml, respectively, according to the Federal Registry as of 2017. I am not sure how this relates to what is actually being tested. In the first and third study the ants are fed sugar water solutions containing 0.5 μg/ml and 0.2 μg/ml triflumezopyrin. How much a.i. do the ants actually consume? It isn’t clear in the methods how Figure 2A and 2B are generated. In a second study, I think the filter paper is treated with these solutions, but it is not clear.

How does this relate to the food tolerances of rice? The methods and material section as written are totally inadequate. It is necessary for the reader to look up each reference, Pan et al. (2017), Huang et al. (2016) and Wang et al. (2015), to be able to understand how the tests and data being generated.

Is any concentration of triflumezopyrim in sugar water toxic to the fire ants? You mention on line 121 that these are sublethal doses. I think the toxicity of triflumezopyrin must be established if you want to actually claim them as sublethal doses.

Validity of the findings

The title is misleading by claiming the effect of residues. At least 2 of studies involved ants consuming sugar water baits.

In the conclusion the authors state that "further studies using field-realistic concentrations of triflumezopyrim are needed in order to support these findings." What are realistic concentrations? They should have been tested in this study.

Additional comments

Title and Line 19: In several of the experiments the ants are feeding on sugar water solutions containing triflumezopyrin. These are not residues.
Line 25-26. Ok. So what does that mean.
Line 51-52. What nontargets insect are impaired?
Line 66. Topical, contact or oral imidacloprid treatment?
Line 166. I really don’t understand this result at all. If triflumezopyrim were toxic, then I could see a decrease in the ability to hold on to a substrate. I don’t see why it should increase.
Lines 191-192. Are RIFA eating the treated rice? I don’t understand what the logic behind these statements.
Line 200. I don’t think paradigms is the correct word here.
Line 213. Why not test the toxicity of triflumezopyrim instead of just assuming that it is less toxic?
Line 236. This study should have used “field-realistic concentrations.”

Reviewer 2 ·

Basic reporting

no comment

Experimental design

no comment

Validity of the findings

no comment

Additional comments

The manuscript “Effects of residues of triflumezopyrim exposure on fitness of red imported fire ant Solenopsis invicta” by Li et al, investigated the chronic impacts of triflumezopyrim to red imported fire ant, Solenopsis invicta after 56 days of exposure from free feeding. The three experimental paradigms which include food consumption and colony growth; active behavior (grasping ability and walking speed) and the aggressiveness were monitored in a 56-day treatment. Their results indicated that 0.5 ug/ml and 0.2ug/ml triflumezopyrin increased individual aggressive index, and not in group aggressive encounters; 0.5 ug/ml triflumezopyrin increased moving speed and grasping ability, while 0.2ug/ml triflumezopyrin only increased grasping ability alone. However, neither 0.5 ug/ml nor 0.2 ug/ml triflumezopyrin had significant impact on S.invicta colony growth and food consumption. For the manuscript, I have several major and minor comments.
Major comments:
1. The red imported fire ant, Solenopsis invict, is a well-known invasive ant species that was introduced into China since 2003. As an eusocial insect, the physiology and behavior of S. invicta in subcolony could change in a 56 day treatment, so it is very important to use the entire colony, not subcolony, for long-term experiments in social insect study. If only workers in the subcolony, growth and food consumption bioassay and active assay were not comparable to the whole colony (include workers, brood, larva, pupa, female, male alate and queen) in natural condition in 56 days treatment.

2. Make sure your result descriptions are consistent:

Abstract L21, Our results showed 0.5 ug/ml not 0.2ug/ml triflumezppyrin increased moving speed and grasping ability of S. invicta.
Result, L165, Grasping rate of workers in triflumezopyrim treatments is higher than that in control.
Result, L173, 0.2ug/ml triflumezopyrim treatment is not significant from either that in 0.5 ug/ml triflumezopyrim or control.
Discussion, L194, Although 0.2 and 0.5 ug/ml triflumezopyrin treatment increased grasping ability, walking speed …..,
Discussion, L205, Our results showed 0.2 and 0.5 ug/ml triflumezopyrin treatment increased grasping ability, walking speed …..,

3. How did you conclude that triflumezopyrin did not have any negative impact on the fitness level of S. invicta at 0.5 ug/ml and 0.2 ug/ml exposure regimes? Three paradigms which related to the fitness of a fire ant colony were chosen in this study, however, two paradigms have significant impacts on S.invicta subcolony.

4. It is the best to generate a separate table for the results and statistical analyses for food consumption result.

5. What is the social form of the fire ant, Solenopsis invict, in your study, which needs to be added in MS.

6. Paper needs to go over thoroughly to optimize sentence structure
Minor comments:

Spelling errors in L101, L129 and L134.

---

## Round 0.2 · Minor Revisions

Dear Dr. Li and colleagues:

Thanks for revising your manuscript. The reviewers are very satisfied with your revision (as am I). Great! However, there are a couple minor issues raised by the reviewers. Please address this ASAP so we may move towards acceptance of your work.

Best,

-joe

Reviewer 1 ·

Basic reporting

no comment

Experimental design

no comment

Validity of the findings

no comment

Additional comments

The authors have done a nice job of revising the manuscript. I have made a few minor comments along the margin of the revised draft.

Annotated reviews are not available for download in order to protect the identity of reviewers who chose to remain anonymous.

Reviewer 2 ·

Basic reporting

no comment

Experimental design

no comment

Validity of the findings

no comment

Additional comments

The authors followed the reviewer’s comments, made most of changes basically. Especially, they used language editing services to improve the English languages of the manuscripts. Therefore, the current version is suitable for publication with minor revision. I have two major and two minor comments.
Major comments
1. Make sure your result description is correct
Abstract L23-24 Our results showed 0.5 μg/ml not 0.2 μg/ml triflumezopyrim improved the grasping ability of S. invicta, (Line 167-168, your statistical analyses result showed 0.2 μg/ml triflumezopyrim and control, Mann-Whitney test, U=18, P=0.002) and 0.5 μg/ml triflumezopyrim increased their moving speed.

Please Change to “Whiles both 0.5 μg/ml and 0.2 μg/ml triflumezopyrim improved the grasping ability of S. invicta, and 0.5 μg/ml not 0.2 μg/ml triflumezopyrim increased their moving speed. In addition, although ….. ”

2. Statistical analyses result letter was showed in Table 1, but not in table 2, please keep consistent.

Minor comments:
1. Line 190 10 ng or 1.25?? /insect
2. Spelling errors in L67, L128, L133 and L139.

---

## Round 0.3 · accepted · Accept

Dear Dr. Li and colleagues:

Thanks for re-submitting your revised manuscript to PeerJ, and for addressing the concerns raised by the reviewer. I now believe that your manuscript is suitable for publication. Congratulations! I look forward to seeing this work in print, and I anticipate it being an important resource for research communities studying control measures for the red imported fire ant.

Thanks again for choosing PeerJ to publish such important work.

-joe